# Fecal Metabolomics Reveals the Foraging Strategies of Giant Pandas for Different Parts of Bamboo

**DOI:** 10.3390/ani13081278

**Published:** 2023-04-07

**Authors:** Zheng Yan, Qin Xu, Ying Yao, James Ayala, Rong Hou, Hairui Wang

**Affiliations:** 1Chengdu Research Base of Giant Panda Breeding, Chengdu 610081, China; 2Sichuan Key Laboratory of Conservation Biology for Endangered Wildlife, Chengdu 610081, China; 3Sichuan Academy of Giant Panda, Chengdu 610081, China

**Keywords:** giant panda, metabolites, conservation, dietary fiber, nutrition, gut microbiota

## Abstract

**Simple Summary:**

Climate change can have a negative impact on the food sources of wild animals, including giant pandas, who primarily rely on bamboo. To better understand the foraging strategy of giant pandas and aid in their conservation, metabolomic methods were used to analyze fecal metabolites and correlate them with gut microbiota. The results indicate that the sugar content in giant panda feces is higher when they consume bamboo culm (the hollow, jointed stem) with high fiber content and that *Streptococcus* bacteria are positively correlated with glucose and acetic acid content, both of which are related to fiber. These results suggest that the foraging strategy of giant pandas is based on the nutritional composition of different bamboo parts, and therefore, enriching their habitat with bamboo species is essential to achieve their natural foraging strategy to obtain adequate nutrition.

**Abstract:**

Climate change-induced food shortages pose major threats to wildlife conservation, and the exclusive reliance of giant pandas on bamboo makes them particularly vulnerable. The aim of this study was to provide insight into the reasons for the foraging strategies of giant pandas to selectively forage for different bamboo parts (bamboo shoot, culm, and leaf) during different seasons. This study used a metabolomic approach to analyze the fecal metabolites of giant pandas and conducted a correlation analysis with their gut microbiota. The results indicate that the fecal metabolites of giant pandas differ significantly depending on the bamboo parts they forage on, with higher sugar content observed when they consume bamboo culm with high fiber content. By functional annotation, culm group metabolites were enriched in the galactose metabolic pathway, while shoot group metabolites were enriched in the phenylalanine, tyrosine and tryptophan biosynthesis pathways. Moreover, *Streptococcus* showed a significant positive correlation with glucose and acetic acid content. Therefore, the foraging strategy of giant pandas is based on the ability to utilize the nutrient content of different bamboo parts. Captive feeding and habitat construction should enrich bamboo species to allow them to express their natural foraging strategies and improve their welfare and reproductive status.

## 1. Introduction

As climate change accelerates and ecological fragmentation increases worldwide, it is becoming more difficult for the natural habitat of numerous species to meet foraging needs [1]. A major component of the in-situ conservation of vulnerable and endangered species, such as the iconic giant panda (*Ailuropoda melanoleuca*), is whether or not they have accessibility to a stable food supply in the wild [2,3]. Due to giant pandas exclusively feeding on bamboo and the potential food crisis caused by the death of large numbers of bamboo after flowering, further exacerbated by habitat loss and fragmentation [4,5], a controversy arose amongst conservationists concerning whether the future of the giant panda has reached an evolutionary cul-de-sac [6]. Given the giant panda’s dependency on bamboo, which has both low nutrient and caloric content, understanding the nutritional intake and utilization of this dietary specialist will provide insight into the adaptive evolution of the animal and ultimately facilitate both the in-situ and ex-situ conservation of the species [7,8,9].

The factors responsible for the evolution of giant pandas from a primarily carnivorous ancestor to an obligate bamboo forager consist of both environmental constraints and genetic changes [1,6]. The adaptations in its morphology, behavior, and diet have enabled it to survive during its eight million years of evolutionary history [10,11,12,13]. Wild giant pandas possess a unique foraging strategy, choosing to feed on different anatomical parts of bamboo during different seasons, which include shoot, leaf, and culm [14]. The nutritional mechanisms underlying this foraging strategy of giant pandas requires further investigation, however, previous research has shown that these three parts of bamboo contain distinct nutritional differences [4]. Compared with the culm, bamboo shoots and leaves both have higher protein and fat, while the culm has a higher fiber content [4]. Although the giant panda has evolved to feed exclusively on bamboo, the species maintains the simple gastrointestinal tract of a carnivore, while a deficiency of cellulase enzymes further reduces their ability to digest fiber in bamboo [15]. As a result, giant pandas defecate frequently, and the main component of their feces is undigested bamboo. As a result of these traits, giant pandas are an ideal species to study using fecal metabolite analysis.

Surveys of fecal metabolites provide noninvasive evidence of food digestion and utilization, as well as a closer reflection of the metabolic interactions of the mammalian-microbiome system [16]. Fecal metabolites are subject to complex interactions between host nutritional intake, genomic, and behavioral changes, and thus, fecal metabolic mapping, as an important instrument for investigating the associations between microbiome composition, host phenotypes, and genetically complex traits, is already well developed [17]. Evidence increasingly indicates that there is a correlation between metabolomics and the gut microbiome and that microbial metabolism produces a multitude of chemicals, hormones, and vitamins that help maintain host metabolic homeostasis [18]. Mammalian intestines, with diverse microbial communities, maintain an extensive metabolome pool that differs from but complements the action of mammalian enzymes in the liver and intestinal mucosa, including functions critical to host digestion [19,20].

Considerable attention exists to the potential impact of diet among various factors affecting gut bacteria [21,22]. Significant differences in the composition and function of the gut microbiota of giant pandas caused by differences in the intake of nutrients when feeding on different anatomical parts of bamboo were demonstrated both in captivity [23] and in the wild [24]. Furthermore, significant changes in body weight, blood biochemical parameters, and serum metabolites were observed in captive giant pandas provisioned with different parts of bamboo [4]. Given that the dietary intake of different bamboo parts significantly influences the digestion and health of giant pandas, the aim of employing fecal metabolic research is to investigate the underlying mechanisms and influences of this strategy. The objective of this study was to investigate how the fecal metabolites of giant pandas were influenced by three distinct diets, each containing a specific anatomical part of bamboo. To achieve this, we first utilized untargeted metabolomics to explore the overall fecal metabolite profile of the pandas, allowing us to cover a broad range of biomolecules. We then conducted a more thorough analysis using targeted metabolomics, specifically focusing on energy-related metabolites, amino acids, and short-chain fatty acids (SCFAs) in order to provide a comprehensive qualitative and quantitative understanding [25]. In addition, the correlation between fecal metabolites, blood metabolites, and gut microbiota was investigated in giant pandas consuming different parts of bamboo to better understand how the experimental subject adapts to different dietary conditions. This metabolomic study builds on previous research on the effects of nutrient intake from bamboo on blood metabolites [4] and gut microbiota [23] and provides an overall perspective of nutrient utilization of this vulnerable species that specializes in bamboo consumption.

## 2. Materials and Methods

### 2.1. Animals and Sample Collection

All fecal samples were collected from a group of 19 healthy adult mixed-sexed giant pandas (age 7–18, median age 10, 10 males, 9 females) housed at the Chengdu Research Base of Giant Panda Breeding (CRBGPB), Sichuan, China, following the methods described in our previous study [23]. In brief, fecal samples were collected from individuals that were fed a continuous diet consisting of a single bamboo part, denoted as shoot, leaf, and culm, for at least 20 days. All subjects were monitored by CRBGPB veterinary staff during the study period, and no medical treatments were performed that could affect the analysis of the samples, including the addition of antibiotics. Fecal samples were collected no later than 10 min after defecation. The outer layer of the sample in contact with the ground was removed to avoid environmental contamination and then packed and stored at −80 °C within 20 min at the CRBGPB. Twenty-seven fecal samples were collected from 19 individuals according to the feeding of different bamboo parts at different times of the year (Appendix A), with a total of 9 samples collected from each group. The sex ratio of the shoot and leaf groups was 6 males and 3 females, and the culm group was 3 males and 6 females. All samples were assayed for untargeted metabolites and targeted metabolites, including energy-related metabolites, amino acids, and short-chain fatty acids (SCFAs).

### 2.2. Sample Pretreatment

A standard fecal sample of 60 mg was mixed well with 200 µL of ultrapure water, and then 800 µL of cold methanol: acetonitrile (1:1, *v*/*v*) was added. Vibrator (QT1, QiTe, Shanghai, China) and ultrasonic cell disruptor (JY92-II, SCIENTZ, Ningbo, China) were used for vortexing and sonication at low temperatures for 30 min. To precipitate the proteins, samples were incubated at −20 °C for 1 h and then centrifuged at 13,000 rpm and 4 °C for 15 min. The clear supernatant was removed and freeze-dried, then stored at −80 °C. In parallel with the preparation of the test samples, quality control (QC) samples were prepared. QC samples were made by mixing equal volumes of each sample.

### 2.3. Untargeted Metabolomics

The untargeted metabolite profiles of giant panda fecal samples were performed by hydrophilic interaction liquid chromatography-ultra performance liquid chromatography-triple time of flight mass spectrometry (HILIC UHPLC-Q-TOF MS). The liquid chromatography-mass spectrometry (LC-MS) analysis was conducted using a UHPLC system (1290 series, Agilent Technologies, Santa Clara, CA, USA) coupled to a triple time-of-flight (TOF) mass spectrometer (AB/Sciex, Foster City, CA, USA) equipped with an electron spray ionization (ESI) source in positive (ESI^+^) and negative (ESI^−^) ion modes.

Liquid chromatography conditions. The HILIC chromatography column (Waters, ACQUITY UPLC BEH Amide 1.7 µm, 2.1 mm × 100 mm column) temperature was maintained at 25 °C. The flow rate was set at 0.3 mL/min, and the sample injection volume was 2 μL. The mobile phase composition A consisted of water and 25 mM ammonium acetate plus 25 mM ammonia, and B was acetonitrile. The gradient elution program was 95% B from 0 to 1 min, B linearly varied from 95% to 65% from 1 to 14 min, B linearly varied from 65% to 40% from 14 to 16 min, B maintained at 40% from 16 to 18 min, B linearly varied from 40% to 95% from 18 to 18.1 min, and B maintained at 95% from 18.1 to 23 min. The samples were placed at 4 °C throughout the analysis. To avoid the effects of fluctuations in the detection signal of the instrument, the samples were analyzed continuously in random order. QC samples were inserted in the sample queue to monitor and evaluate the stability of the system and the reliability of the experimental data.

Q-TOF mass spectrometry conditions. The mass spectrometer (MS) used was a triple TOF 5600 system (AB/Sciex, Foster City, CA, USA) equipped with an ESI source. ESI source conditions were set as followings: Ion Source Gas1, 60 psi; Ion Source Gas2, 60 psi; curtain gas, 30 psi; source temperature, 600 °C; IonSapary voltage floating (ISVF), ±5000 V; TOF MS scan range, m/z of 60–1000 Da; product ion scan m/z range, 25–1000 Da; TOF MS scan accumulation time, 0.2 s per spectra; product ion scan accumulation time, 0.05 s per spectra; MS/MS data were acquired in the information-dependent acquisition (IDA) mode and high sensitivity modes. Declustering potential (DP): ±60 V (ESI^+^ and ESI^−^); collision energy, 35 ± 15 eV. Settings of IDA were as follows: exclude isotopes within 4 Da, candidate ions to monitor per cycle, 6.

Metabolite ion peaks analysis. The raw data were converted to mzXML format by ProteoWizard (version 3.0.20103, ProteoWizard software foundation, Los Angeles, CA, USA), and then the XCMS program was used for peak alignment, retention time correction, and extraction of peak areas. The metabolite structure identification was performed using exact mass number matching (<25 ppm) and secondary spectra matched to search the database [26,27]. Patterns were identified by applying the software SIMCA-P (Umetrics, Umea, Sweden), and the data were preprocessed by Pareto-scaling and used for subsequent multidimensional statistical analysis.

### 2.4. Targeted Metabolomics

Three targeted metabolites were the focus of this study. Energy-targeted metabolomics involves 29 important metabolites in the glycolytic pathway, tricarboxylic acid (TCA) cycle pathway, and oxidative phosphorylation pathway. Amino acid-targeted metabolites include amino acids and their derivatives. Short-chain fatty acid (SCFA)-targeted metabolites include acetic acid, propionate, butyrate, isobutyric acid, valeric acid, isovaleric acid, and caproic acid. Energy-related and amino acid-related metabolites were detected by ultra-performance liquid chromatography-mass spectrometry/mass spectrometry (UHPLC-MS/MS) based on multiple reaction monitoring (MRM) targeted metabolomics [28]. The MRM method provides targeted and specific detection and analysis of specific metabolite groups concerning standard metabolites and provides absolute quantitative results of target metabolites with high specificity, sensitivity, and accuracy [29]. Metabolites of SCFAs were detected and selected by gas chromatography-mass spectrometry (GC/MS) based on the selected ion monitoring (SIM) method. SIM is primarily used to detect several m/z characteristic ions specific to the target and ignore other m/z fragment ions for a defined period. SIM technique focuses the limited scan time on the detection of specific characteristic ions, enhancing the target signal and improving the sensitivity [30].

Energy metabolites targeted detection. An Agilent 1290 Infinity LC ultra-performance liquid chromatography system (Agilent, Santa Clara, CA, USA) was used for the analysis. The sample was placed in a 4 °C autosampler with a HILIC chromatography column (Waters, ACQUITY UPLC BEH Amide 1.7 µm, 2.1 mm × 100 mm column) at 45 °C, a flow rate of 300 μL/min, and an injection volume of 4 μL. The relevant liquid phase gradients were as follows: 0–18 min, liquid B varied linearly from 90% to 40%; 18–18.1 min, liquid B varied linearly from 40% to 90%; 18.1–23 min, liquid B was maintained at 90%. Mass spectrometry was performed in negative ion mode using a 5500 QTRAP mass spectrometer (AB SCIEX). The 5500 QTRAP ESI source conditions were as follows: source temperature 450 °C, ion Source Gas1: 45, Ion Source Gas2: 45, Curtain gas: 30, ISVF: −4500 V. The ion pairs to be measured were detected in MRM mode.

Amino acid and derivatives targeted detection. Agilent 1290 Infinity LC ultra-performance liquid chromatography system was used for the analysis. Mobile phase A consisted of 25 mM ammonium formate + 0.08% Formic acid (FA) in H_2_O, and mobile phase B consisted of 0.1% FA in acetonitrile. All samples were kept at 4 °C in an automatic sampler throughout the analysis, the column (Zic-HILIC 3.5 µm, 2.1 mm × 150 mm column) temperature was kept at 40 °C, the flow rate was 250 μL/min, and the injection volume was 2 μL. Gradient conditions were as follows: start with a linear variation from 90% B to 70% B in 12 min, B changed linearly from 70% to 50% at 12 to 18 min, from 50% to 40% at 18 to 25 min, from 40% to 90% at 30 to 30.1 min, and was maintained at 90% at 30.1 to 37 min. Mass spectrometry was performed using a 5500 QTRAP mass spectrometer (AB SCIEX) in positive ion mode. The 5500 QTRAP ESI source conditions were as follows: source temperature 500 °C, ion Source Gas1: 40, Ion Source Gas2: 40, CUR: 30, ISVF: 5500 V. The ion pairs to be measured were detected in MRM mode.

Short-chain fatty acid targeted detection. The standards were prepared with ether into nine concentration gradients of 0.05 μg/mL, 0.1 μg/mL, 0.2 μg/mL, 0.5 μg/mL, 1 μg/mL, 5 μg/mL, 10 μg/mL, 25 μg/mL, and 50 μg/mL for GC-MS detection, with an injection volume of 1 μL and a splitting ratio of 10:1. Fifty mg of the sample was added to 50 μL of 15% phosphoric acid, then 5 μg/mL of internal standard (isocaproic acid) solution was added to 150 μL and mixed well, centrifuged at 12,000 rpm at 4 °C for 10 min, and the supernatant was extracted into GC-MS with an injection volume of 1 μL and a splitting ratio of 10:1.

An Agilent HP-INNOWAX capillary column (30 m × 0.25 mm ID × 0.25 μm) was used for the separation. The initial temperature was 90 °C, then 10 °C/min to 120 °C, then 5 °C/min to 150 °C, and finally 25 °C/min to 250 °C for 2 min. Helium was used as a carrier gas at a flow rate of 1.0 mL/min. An Agilent 7890A/5975C gas-mass spectrometer was used for mass spectrometry analysis. The inlet temperature was 250 °C, the ion source temperature was 230 °C, the transmission line temperature was 250 °C, and the quadrupole temperature at 150 °C. The electron bombardment ionization (EI) source was set at a full sweep, the SIM was set in scan mode, and the electron energy was set at 70 eV.

QC samples were randomly positioned in the column during the analytical sequence to monitor the precision and stability of the method during the operation. Peak areas and retention times for energy-related metabolites and amino acids were extracted by Multiquant software (version 3.0, Applied Biosystems, Foster city, CA, USA), and retention times were corrected with standards for metabolite identification. Peak areas and retention times for short-chain fatty acid assays were extracted by MSD ChemStation software (version C.01.08, Applied Biosystems, Foster city, CA, USA). Finally, the standard curve was plotted, and the short-chain fatty acid content was calculated.

### 2.5. KEGG Pathway Enrichment Analysis

Differential metabolites (biomarkers) were submitted to the Kyoto Encyclopedia of Genes and Genomes (KEGG) website (https://www.kegg.jp/kegg/pathway.html accessed on 10 February 2022) for related pathway analysis. The function of Biomarkers on the primary and secondary pathways of KEGG was obtained [31].

### 2.6. Statistical Data Analysis

Data were preprocessed by Pareto scaling and subjected to multivariate statistical analyses by R software (version 4.2.0, R foundation for statistical computing, Vienna, Austria), including unsupervised principal component analysis (PCA), supervised partial least squares discriminant analysis (PLS-DA), and orthogonal partial least squares discriminant analysis (OPLS-DA). The importance of the biomarkers was ranked using a variable importance prediction (VIP) score >1 in the OPLS-DA model. For univariate analysis, candidate-specific biomarkers were identified using Student’s *t*-test statistics for comparison of means between two groups, and *p* < 0.05 was considered to be statistically significant. The fecal metabolite abundance data for the enrichment heatmap were log-transformed to make the results clearly presented. The abscissa in the heatmap was the log value of log_2_ of the metabolite abundance.

Gut microbial genomic and fecal metabolite detection from the same fecal samples of giant pandas ensured the reliability of the data obtained for correlation. Spearman rank correlation coefficients were calculated between all metabolite intensities and the relative abundances of all bacterial genera and encoded enzymes among all samples from all giant pandas using the “cor” function in R software (4.2.0). Metagenome data and 16S rRNA gene data of gut microbiota were derived from previous research [23]. The data are available from the Sequence Read Archive under BioProject PRJNA433781. To determine whether fecal samples with similar bacterial compositions also had similar metabolite profiles, a Mantel test was performed, and the plot was generated using R software (4.2.0).

## 3. Results

### 3.1. Metabolomic Profiles

HILIC UHPLC-Q-TOF/MS-based untargeted metabolomics assay was used to obtain an overall outline of fecal metabolites in giant pandas. The total ion chromatograms (TICs) of quality control (QC) samples were compared for spectral overlap and were made. The response strength and retention time of each chromatographic peak overlapped, indicating that the variation caused by instrumental error was small during the entire experiment (Appendix A). By partial least squares discrimination analysis (PLS-DA), QC samples were closely clustered under the positive and negative ion modes (Figure 1a,b), indicating that the project experiment had high repeatability. The culm, leaf, and shoot groups can be clustered separately under the positive and negative ion modes (Figure 1a,b).

A total of 5675 metabolic peaks were detected, including 2854 positive patterns and 2821 negative patterns. Orthogonal partial least squares discriminant analysis (OPLS-DA) of fecal metabolites in giant pandas under different dietary conditions to obtain a higher level of group separation and a better understanding of the variables responsible for group classification, the values of the R^2^Y and Q^2^ intercept indicate the robustness of the models and thus show a low risk of overfitting and reliability (R^2^Y > 0.98, Q^2^ > 0.96, Appendix A for details). The metabolic peaks with variable importance in the projection (VIP) >1 and *p* < 0.05 between every two groups were selected to be significantly different metabolites between groups (biomarkers), and a total of 114 biomarkers were identified (Appendix A). These metabolites were mainly classified into amino acids, amino acid metabolism, bile acids, energy metabolism, fatty acids, nucleotide metabolism, sugars, and vitamins. The culm group showed lower amino acid, bile acid, fatty acid, and vitamin content and higher sugar content, including alpha-D-glucose, fructose, mannose, stachyose, raffinose, and sucrose, than the other diet groups. Among the differential metabolites involved in energy metabolism, the content of lactic acid was higher in the shoot group, the content of succinic acid was lower, and the flavin mononucleotide (FMN) was the lowest in the shoot group (Figure 1c).

The metabolites with significant differences (biomarkers, variable importance prediction (VIP) score > 1 and *p* < 0.05) between groups were submitted to the Kyoto Encyclopedia of Genes and Genomes (KEGG) website for relevant pathway analysis, which showed that biomarkers between leaf and culm groups, between leaf and shoot groups, and between shoot and culm groups involved 30, 8, and 39 KEGG pathways, respectively (Appendix A). In addition, the biomarkers between culm and shoot groups and between culm and leaf groups involved 26 common pathways, of which culm groups had a significantly lower abundance of biomarkers in protein digestion and absorption, aminoacyl-tRNA biosynthesis, phenylalanine, tyrosine, and tryptophan biosynthesis pathways, and in the biosynthesis of unsaturated fatty acids and linoleic acid metabolism and mineral absorption pathways. However, the culm group metabolites were enriched in the galactose metabolic pathway. The shoot group exhibited more metabolites in phenylalanine, tyrosine and tryptophan biosynthesis compared to the other two groups, including phenylalanine, quinate, shikimate, tyrosine, and anthranilic acid (Vitamin L1). The abundance of metabolites-related pathways in the bamboo leaf group was not significantly different compared to the other two groups simultaneously. However, compared to the bamboo shoot group, the leaf group had significantly lower metabolites associated with the starch and sucrose metabolic pathway (including sucrose, cellobiose, and isomaltose), galactose metabolic pathway (including galactitol, sucrose, raffinose, and stachyose), pyrimidine metabolic pathway (including thymidine, uracil, uridine, thymine, phenylalanine, and tyrosine), and tryptophan biosynthetic pathway (including quinine, shikimate, and anthranilic acid (vitamin L1)). The top 10 most significant enrichment pathways were shown in Figure 2.

### 3.2. Targeted Metabolomics

Targeted metabolomics was used to more precisely explore the effects of different nutrient intakes on the fecal metabolites of giant pandas. We focused on metabolites in terms of energy, amino acid, and short-chain fatty acids. The quality control samples of the three experiments were subjected to PLS-DA, QC samples were clustered together, and the system stability was relatively stable and reliable during the whole experiment. More specific exploration of metabolites resulted in the overlap between groups (Figure 3a–c).

#### 3.2.1. Energy-Targeted Metabolites

A total of 17 energy-related metabolites were detected in the glycolytic pathway, tricarboxylic acid (TCA) cycle pathway, and oxidative phosphorylation pathway. The bamboo leaf group had a higher abundance of metabolites associated with energy metabolic functions at the overall level (Figure 3d). Fold change analysis showed that nine metabolites of these pathways were significantly different between groups (*p* < 0.05, Appendix A). Pyruvate, aconitate, and lactate levels were higher in the leaf group compared to the shoot and culm groups, while oxaloacetate levels were lower in the leaf group. The malate level was higher in the leaf group than in the shoot group. The highest level of flavin mononucleotide (FMN) was in the culm group, followed by the leaf group, and the lowest in the shoot group. Nicotinamide adenine dinucleotide+ (NAD+) and adenosine monophosphate (AMP) were higher in the culm group than in the shoot group. The cyclic Adenosine monophosphate (cyclic-AMP) level was lower in the shoot group than in the other two groups.

#### 3.2.2. Amino Acid

A total of 28 amino acids and their derivatives were detected in giant panda feces, and the difference multiplicity analysis showed that 23 of them were significantly different among the groups (*p* < 0.05, Appendix A). The contents of mostly amino acids were lower in the culm group than in the shoot and leaf groups (Figure 3e), including aspartate, methionine, proline, threonine, alanine/sarcosine, citrulline, ornithine, glycine, lysine, serine, phenylalanine, tyrosine, leucine, isoleucine, histidine, valine, glutamate, tryptophan, and glutamine. The content of arginine was higher in the leaf group than in the shoot and culm groups. The content of hydroxyproline was higher in the shoot group than in the leaf and culm groups. Cystine content was higher in the leaf group than in the culm group. Asparagine content was higher in the shoot group than in the culm group.

#### 3.2.3. Short-Chain Fatty Acids

Seven short-chain fatty acids (SCFAs) were detected, including acetic acid, propionic acid, isobutyric acid, butyric acid, isovaleric acid, valeric acid, and hexenoic acid. Except for acetic acid, propionic acid and caproic acid, the other metabolites had the highest levels in the bamboo leaf group (Figure 3f and Appendix A). Acetic acid content occupied more than 90% of the detected SCFAs and was significantly higher in the bamboo culm and bamboo leaf groups than in the bamboo shoot group. The content of propionic and caproic acid was significantly lower in the bamboo culm group than in the other groups. The contents of butyric acid, isobutyric acid, isovaleric acid, and valeric acid were significantly higher in the leaf group than in the other two groups. However, at the overall level of SCFAs, no significant difference was found between the culm and leaf groups, and both were higher than the bamboo shoot group (Appendix A).

### 3.3. Microbiota–Metabolome Association

The diet of different bamboo parts also significantly altered the composition of the gut microbiota of giant pandas, with representative gut microbial communities for each feeding period. To investigate the correlation between gut microbiota and the fecal metabolites in giant pandas, spearman correlation analysis between the 16S rRNA genes of the gut microbiota and untargeted metabolites, amino acids, energy metabolites, and short-chain fatty acids (SCFAs) was performed. The 114 untargeted biomarkers were found to be significantly correlated with 155 microbial genera (Appendix A). A total of 28 microbes were significantly correlated with glucose (*p* < 0.05), of which eight were positively correlated, with the correlation coefficients in descending order of *Streptococcus*, *Sphingomona*s (spearman rank correlation coefficient, rho > 0.5 above), *Bifidobacterium*, *Faecalitalea*, *Corynebacterium*, *Rhodanobacter*, *Weissella*, and *Rhodococcus* (rho > 0.4 above).

Thirty-nine bacterial genera in six phyla were significantly associated with fifty-two metabolites detected by three targeted metabolomics (Figure 4). Thirty-nine genus-level gut microbes were classified into six phylum levels. *Streptococcus* was negatively correlated with nineteen amino acids and propionic acid and positively correlated with acetic acid and flavin mononucleotide. *Escherichia-Shigella* was positively correlated with six short-chain fatty acids, five energy-related metabolites, and seven amino acids and negatively correlated with asparagine. *Lactococcus*, *Aeromonas*, *Cellulosilyticum,* and *Cetobacterium* were positively correlated with cysteine, histidine, proline, hydroxyproline, glycine, asparagine, ornithine, citrulline and threonine, and negatively correlated with acetic acid, flavin mononucleotide, and fumarate.

In addition, we correlated the functional genes of gut microbiota with the fecal metabolites. Seven carbohydrate-related metabolites were significantly correlated with four microbial enzyme genes, with isomaltose and stachyose significantly and positively correlated with cellulose 1, 4-beta-cellobiosidase (EC 3.2.1.91). Sucrose and fructose had a significant positive correlation with bata-glucosidase (EC 3.2.1.21) (Figure 5a). Six enzyme genes of the gut microbiota involved in energy-related metabolic pathways with the highest relative abundance in the bamboo leaf group were positively correlated with lactate, GMP, pyruvate, aconitate, isocitrate, and NAD, and the relative abundance of these metabolites was also the highest in the bamboo leaf group (Figure 5b).

## 4. Discussion

In 2016 the giant panda was downgraded from endangered to threatened by the International Union for the Conservation of Nature [32]. However, as a bamboo specialists, giant pandas are still vulnerable to extinction as a result of habitat fragmentation and the potential effects of climate change as well as the poorly understood phenomena of bamboo die-offs [7,8,9]. Understanding the unique foraging strategy of this elusive species of bear and how it relates to their nutritional balance is a necessary component of both the in-situ and ex-situ conservation of the species. To improve the understanding of the giant panda foraging strategy, this study compared the utilization of different nutrients by giant pandas in bamboo culm, left, and shoot, respectively, using a fecal metabolomics approach combined with gut microbial data [23]. The results showed significant differences in fecal metabolic profiles among the three bamboo part groups and significant correlations between fecal microbiota and metabolites, thus revealing the digestion and utilization of different bamboo parts by giant pandas and their foraging strategies.

During high-fiber bamboo culm feeding, we found higher sugar concentrations in the fecal metabolites of the giant panda. Glucose levels in the serum of giant pandas were also higher when bamboo was consumed [4]. During this period, the gut microbiota (GM) of giant pandas exhibit a high abundance of glycoside hydrolases and carbohydrate esterases, which are involved in the degradation of polysaccharides [23]. The GM of giant pandas performs an essential function in the digestion and utilization of polysaccharides. The GM of the bamboo culm group also possessed more genes encoding glucokinase, the key enzyme in the glycolysis pathway that catalyzes the glucose-6-phosphate reaction [23]. Thus, the giant panda GM enabled the utilization of fiber for energy production when consuming bamboo culm. Moreover, giant pandas prefer to forage for bamboo culm with higher mono- and polysaccharides content in spring [33], which may be the foraging strategies based on their ability to ingest nutrients from the fiber.

Dietary fiber also serves as the most important substrate for the production of short-chain fatty acids (SCFAs), which are absorbed in the colon after fermentation by GM [34]. The fiber content of bamboo culm and leaves were significantly higher than that of bamboo shoots, and the SCFAs of giant panda feces followed the above trend with the fiber content, as was expected. However, the fiber content of bamboo culm was significantly higher than that of bamboo leaves, and there was no difference in SCFAs content between these two bamboo parts when they were consumed. Previous research suggests that excessive fiber content would no longer increase SCFAs levels [35], whereas protein and amino acid degradation is an easily overlooked way of SCFAs production [36]. Several amino acids released by proteins in the colon are precursors for the synthesis of SCFAs, such as glycine, alanine, threonine, glutamic acid, lysine, and aspartic acid used by microbiota to produce acetate [37]. Bamboo leaves were significantly higher in protein and amino acids than bamboo culm [4] and were higher in isobutyrate and valerate [38], these substances were also higher in the fecal metabolites of the bamboo leaf group. Therefore, bamboo leaves contained more abundant substrates for the production of SCFAs.

Acetic acid accounted for more than 90% of all SCFAs in giant panda feces and was highest in the feces of the bamboo culm group. Acetic acid represents the most important metabolite produced after fiber fermentation and is also an essential substance for the synthesis of cholesterol, most of which enters the bloodstream and is eventually metabolized by the liver to provide energy to the surrounding tissues [34]. Hence, acetic acid is a vital element for the body to absorb energy from carbohydrates hardly utilized [39]. In addition, *bifidobacteria* are acetic acid-producing bacteria, and a fiber-rich diet also promotes *bifidobacteria* production [40]. The abundance of *bifidobacteria* in the giant panda’s feces was also highest when consuming bamboo culm [23], and additionally, here the role of GM in fiber utilization in giant pandas was also demonstrated. Additionally, butyric acid was the highest in the feces of the bamboo leaf group, and the terminal enzyme gene, acetate CoA-transferase, in the main butyric acid synthesis pathway in the GM of hindgut fermenting mammals was highest in giant pandas during bamboo leaf feeding [23,41]. Butyric acid is known to inhibit intestinal inflammation, the development of intestinal cancer cells, and enhance intestinal defenses [42]. Therefore, the enrichment of butyric acid contributes to the maintenance of intestinal health during the period of bamboo leaf foraging.

Energy-related metabolites in giant panda feces differed significantly among the three bamboo parts of the diet, each feeding period exhibited representative metabolites. The metagenome data showed that the GM of giant pandas’ during different feeding periods were dominant in energy-related enzyme genes in the glycolytic pathway, tricarboxylic acid (TCA) cycle pathway, and oxidative phosphorylation pathway, respectively [23], and the apparent digestibility of energy was not significantly different among the different periods [4]. In addition, the amino acid content of giant panda feces was generally low when feeding on bamboo culm, which was related to the low amino acid content in bamboo culm [43,44]. Therefore, at different foraging periods giant pandas utilized different parts of the bamboo rich in nutrients, such as carbohydrates, proteins, and fats, to meet their nutritional needs.

We discovered the correlation between GM and fecal metabolites of giant pandas, as well as the relationship between the genetic function of GM and nutrient digestion and utilization of giant pandas. The GM of giant pandas systematically utilizes the nutrient components of bamboo to obtain essential energy sources from this generally considered low-nutrient bamboo diet. We identified that *streptococcus* was significantly and positively correlated with glucose and acetic acid, which are the main products of cellulose production through hydrolysis and microbial fermentation [45,46]. Furthermore, *streptococcus* had a significantly higher abundance in the high-fiber bamboo culm diet and as well as encoded enzymes involved in fiber fermentation, including glycoside hydrolases, polysaccharide lyases, and carbohydrate esterase [23]. Putatively, *streptococcus* are the primary degraders or keystone species for dietary fiber utilization in giant pandas. Moreover, the highest abundance of genes related to energy metabolism was found in giant panda gut microbes during bamboo leaf feeding [23], which is consistent with the changes in fecal energy-related metabolites in this study. The above evidence suggests that giant pandas utilize the nutrient content of bamboo in a sophisticated manner and that gut microbes contribute considerably.

This study describes in detail the nutritional utilization of the different parts of bamboo which are seasonally selected by giant pandas, providing a scientific basis for formulating the diets of captive giant pandas as part of the ex-situ conservation of the species. Furthermore, it aids in-situ conservation by providing a clearer comprehension of the foraging strategies of wild giant pandas, thus informing wildlife managers which habitat to prioritize for giant panda conservation and which are potential sites for giant panda reintroductions.

## 5. Conclusions

Overall, this study reveals the variations of metabolites in giant pandas during the foraging of different bamboo parts. The fecal short-chain fatty acid content was higher in giant pandas when foraging on both bamboo culm and leaves, which correlated with the fiber and protein composition of bamboo culm and leaves. When feeding on bamboo culm, which had the lowest protein and amino acid content, giant pandas had the lowest fecal amino acid content. However, the higher fiber content in bamboo culm was utilized by gut microbiota, providing nutrition for the panda, and as a result, their fecal and blood glucose were both highest in the bamboo culm group. The consistent variation in the content of energy-related metabolites and the abundance of functional genes of gut microbiota in the feces during different foraging periods indicated that the gut microbiota utilized the higher content of nutrients in different bamboo parts to supply energy to the giant pandas. In addition, we found a strong correlation between fecal metabolite content and gut microbial abundance in giant pandas and hypothesized that *streptococcus* is important for fiber utilization. The results suggest that the foraging strategy of giant pandas is based on the nutritional content of bamboo and that habitat construction requires the enrichment of bamboo species so that pandas can freely select bamboo parts with high nutritional content to achieve a natural foraging strategy.

## Figures and Tables

**Figure 1 animals-13-01278-f001:**
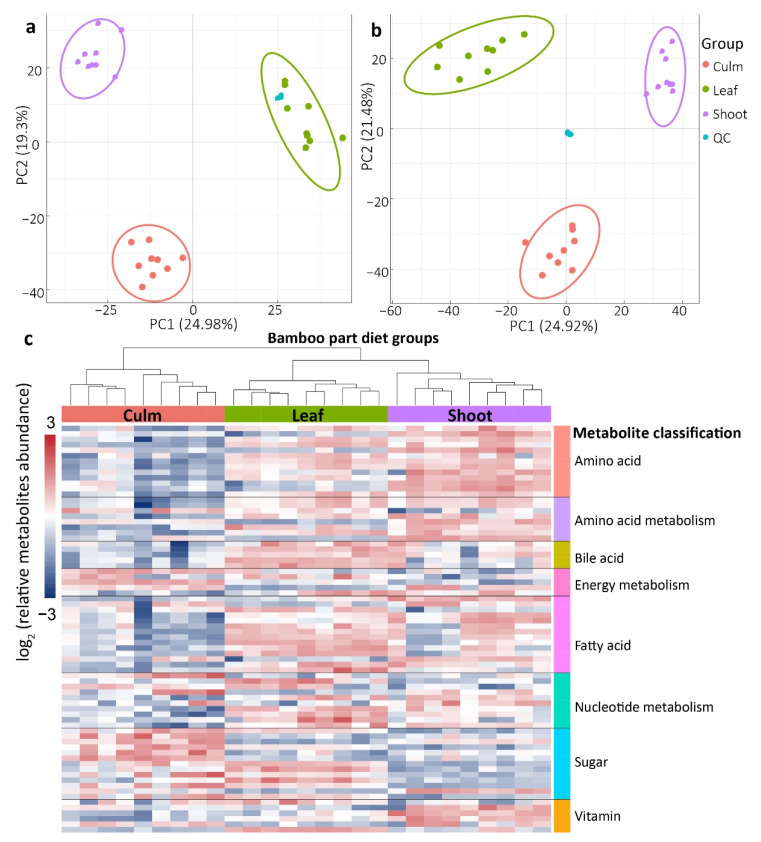
Detected differences in the bamboo culm, leaf, and shoot diet in the overall fecal metabolite profiles of giant pandas by using the non-targeted metabolomics method. The fecal metabolites were separated and clustered into the three bamboo groups in (**a**) positive and (**b**) negative ion pattern PLS-DA score diagram. (**c**) Heatmaps showed that different nutrient intakes caused changes in fecal metabolites. The color indicates the homogenized data, red cells indicate a higher abundance of metabolites, and blue cells indicate a lower abundance. Measured metabolites that could be assigned to a category are shown.

**Figure 2 animals-13-01278-f002:**
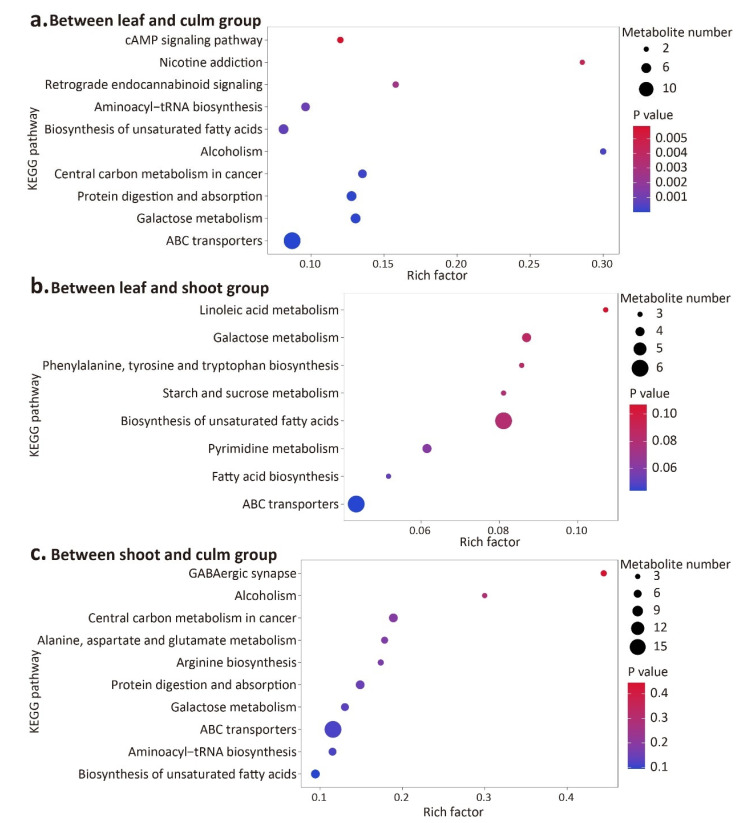
KEGG enrichment analysis bubble map showed that the top 10 metabolic pathways enriched by significant differential metabolites between (**a**) leaf and culm group, (**b**) leaf and shoot group, and (**c**) shoot and culm group. The rich factor is the ratio of the number of significantly different metabolites detected to the number of metabolites annotated in the pathway, with higher rich factor values representing higher levels of enrichment. The size of the point represents the amount of enrichment of significant metabolites in the corresponding metabolic pathway.

**Figure 3 animals-13-01278-f003:**
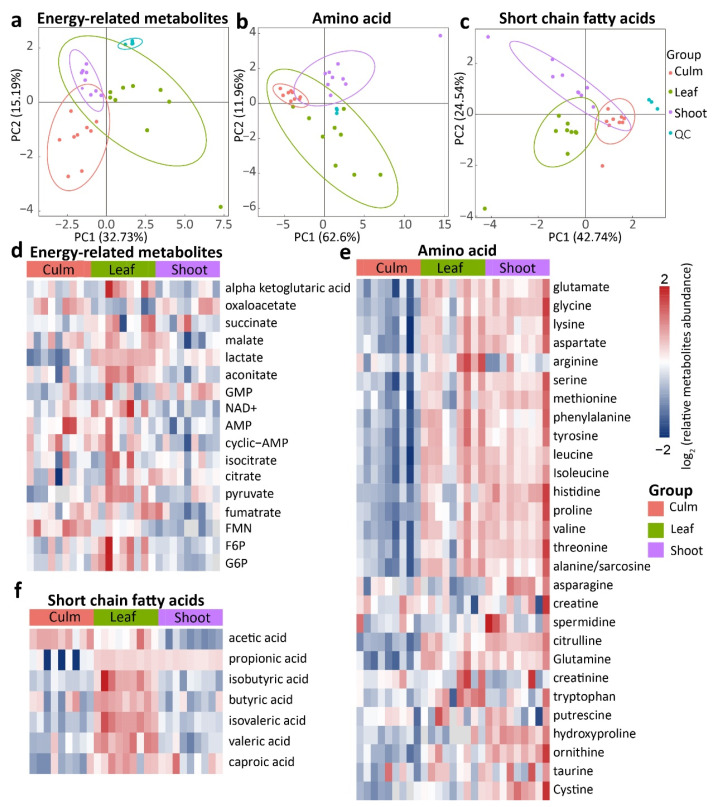
Results of three targeted metabolomics analyses of giant panda fecal samples. (**a**–**c**) Scatter plot of partial least squares discrimination analysis (PLS-DA) scores of fecal metabolites in giant pandas under different dietary conditions. (**d**–**f**) Heatmap of targeted fecal metabolites in giant pandas. Each row in the graph represents a metabolite, and columns of the same color represent a group of samples. Red represents significant up-regulation, blue represents significant down-regulation, and color depth represents the degree of up-regulation.

**Figure 4 animals-13-01278-f004:**
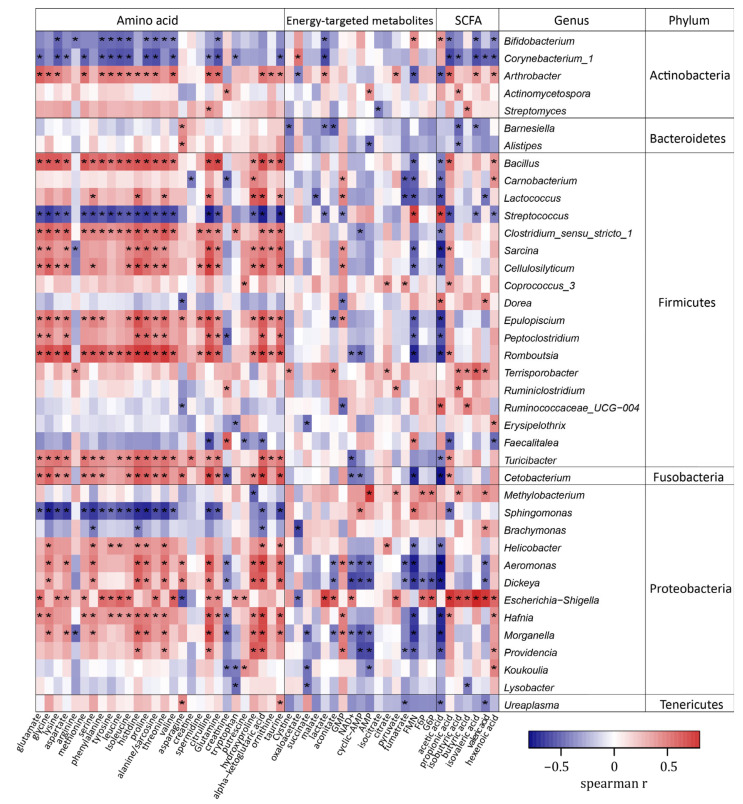
Heatmap of the correlation between gut microbial genera and fecal metabolites in giant pandas. The red and blue squares represent positive and negative correlations between metabolites and gut microbiota, respectively. The * symbols represent *p* < 0.05, ** symbols represent *p* < 0.01.

**Figure 5 animals-13-01278-f005:**
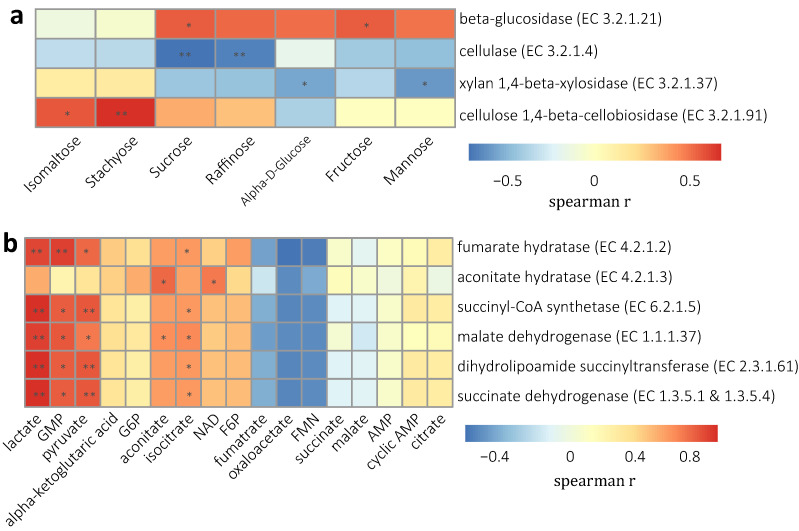
Heatmap of the correlation between functional genes of gut microbiota and fecal metabolites in giant pandas. (**a**) Correlation of carbohydrates-related metabolites with functional genes of gut microbiota. (**b**) Correlation of metabolites involved in energy-related metabolic pathways with functional genes of gut microbiota. The red and blue squares represent positive and negative correlations between metabolites and gut microbiota, respectively. The * symbols represent *p* < 0.05, ** symbols represent *p* < 0.01.

## Data Availability

The datasets used and analyzed during the current study are available from the corresponding authors on reasonable request.

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
