# Peer review of "Fecal Metabolomics Reveals the Foraging Strategies of Giant Pandas for Different Parts of Bamboo"

_animals, 2023, doi:10.3390/ani13081278_

Round 1

Reviewer 1 Report

This manuscript analyzed fecal metabolites of giant pandas and did a correlation analysis with their gut microbiota. This study provides insight into the reasons for the foraging strategies of giant pandas to selectively forage for different bamboo parts (bamboo shoot, culm and leaf). This is a valuable and interesting manuscript. untargeted metabolomics and targeted metabolomics were utilized. Energy related metabolites, amino acids, and short-chain fatty acids were specifically focused. After revision, I suggest it would be accepted in the journal of animals.

Minor:

Line 29 delete “and”

Major:

16S rRNA sequencing data should be provided, at least the data base and number. (line 242-243)

Author Response

Dear reviewer,

I would like to express our gratitude to you for your helpful comments. We have deleted the word "and" in line 29 and provided the information of the 16S rRNA sequencing data base and number as requested in lines 242-243. The data are available from the Sequence Read Archive under BioProject PRJNA433781. Thank you for pointing out the need for this information and making the data more accessible to readers.

Details

(1)Line 29 delete “and”

Response: We have deleted the ‘and’ in line 29, thank you.

(2)16S rRNA sequencing data should be provided, at least the data base and number. (line 242-243)

Response: Yes, the data base and number should be here to let reader available to acquire the date. So we added the information of the dataset in there. The data are available from the Sequence Read Archive under BioProject PRJNA433781.

Thank you again for your help.

Reviewer 2 Report

 Fecal metabolomics reveals the foraging strategies of giant pandas for different parts of bamboo

 By Zheng Yan, Qin Xu, Ying Yao,  James Ayala, Rong Hou and Hairui Wang

General appraisal:

This is an interesting and well conducted study of the giant panda’s diet and the implications of its specialising on different components of bamboo in different seasons. The investigation has both basic and (potential) conservation value. The Methods are clearly and comprehensively described, the Results mostly appropriately analysed and well-presented and the discussion is thoughtful and stimulating. The ms. is well written and mostly very clear and easy to follow.

Detailed comments:

Line 14. Culm (the hollow, jointed stem)

L 17-18. Meaning a bit obscure – clarify.

L 42. Explain why large bamboo blooms cause a potential food crisis for pandas – reason not obvious.

L 53. Is this choice or necessity? Are all three complements available year-round or do they have different seasonal availabilities?

L 67-68. Needs more explanation/elaboration.

L 72. the action of mammalian enzymes

L 103-104. Experimental design unclear at this point. Did all 19 pandas each get each of the treatments and, if so, was the treatment order the same for all individuals OR were there 3 groups, each receiving just one treatment, and if so was the sex ratio approximately similar in all groups?  The reader should not have to refer to a supplementary table to work this out.

L 229 et seq.

(a)   Were data transformed for any of the statistical tests?

(b)   If you do a large number of pairwise tests on a data set, surely some are bound to turn out significant by just by default. Is this a problem here and later?

Fig 4. Networks: this may be the conventional way of presenting such results, but the figures are virtually impossible to decipher, which essentially defeats the purpose!

Author Response

Dear reviewer,

We appreciate the insightful comments from you. We have made revisions to clarify the description of culm (the hollow, jointed stem) in line 14 and to make the meaning of the sentence in lines 17-18 clearer. In line 42, we have explained why large bamboo blooms cause a potential food crisis for pandas by stating that bamboo blooms only once in its lifetime, and most bamboos die after blooming. We have also added the word "wild" in line 53 to clarify that the section describes the foraging strategies of wild giant pandas, which select different parts of the bamboo in different seasons. Additionally, we have provided a more detailed explanation of fecal metabolites in lines 67-68 and made revisions to line 72 according to the suggestion. We also agree with the comment about the experimental design in lines 103-104 and have added a clear description of the design in the manuscript.

Details:

(1) Line 14. Culm (the hollow, jointed stem)

Response: We have added a definition of “Culm” to line 14 to provide a clearer understanding of the parts of the bamboo. Thank you for your comment.

(2) L 17-18. Meaning a bit obscure – clarify.

Response: This sentence really did not express clearly. We have revised the sentence in lines 17-18 as follows: “These results suggest that the foraging strategy of giant pandas is based on the nutritional composition of different bamboo parts, and therefore, enriching their habitat with bamboo species is essential to achieve their natural foraging strategy to obtain adequate nutrition”. Thank you for your suggestion.

(3) L 42. Explain why large bamboo blooms cause a potential food crisis for pandas – reason not obvious.

Response: Bamboo, unlike other plants, blooms only once in its lifetime, and most bamboos die after blooming. We have modified “the potential food crisis caused by large bamboo blooms” to “the potential food crisis caused by the death of large numbers of bamboos after flowering” in line 42 to clarify the reason. Thank you for your comment.

(4) L 53. Is this choice or necessity? Are all three complements available year-round or do they have different seasonal availabilities?

Response: We have added the word “wild” to line 53 to indicate that the section describes the foraging strategies of wild giant pandas, which select different parts of the bamboo in different seasons. Thank you for your suggestion.

(5) L 67-68. Needs more explanation/elaboration.

Response: We have included a description of fecal metabolites in lines 67-68, as follows: “Fecal metabolites are subject to complex interactions between host nutritional intake, genomic, and behavioral changes”. Thank you for your comment.

(6) L 72. the action of mammalian enzymes

Response: We have revised the sentence in line 72 accordingly. Thank you for your suggestion.

(7) L 103-104. Experimental design unclear at this point. Did all 19 pandas each get each of the treatments and, if so, was the treatment order the same for all individuals OR were there 3 groups, each receiving just one treatment, and if so was the sex ratio approximately similar in all groups?  The reader should not have to refer to a supplementary table to work this out.

Response: Admittedly, we did not describe the experimental design clearly in our initial submission. The experiment was conducted in different seasons, and not all animals received all three experimental groups; rather, there were three feeding groups. Therefore, we added the following description: “Twenty-seven fecal samples were collected from 19 individuals according to the feeding of different bamboo parts at different times of the year (Table S1), with a total of 9 samples collected from each group. The sex ratio of the shoot and leaf groups was 6 males and 3 females, and the culm group was 3 males and 6 females”. After modification, we believe that our experimental design is now expressed more clearly, and we appreciate your suggestion”.

(8) L 229 et seq.

(a)   Were data transformed for any of the statistical tests?

Response: We log-transformed the metabolite abundance data when we created the heatmap of fecal metabolite enrichment to make the results clearer. The abscissa in the heatmap was the log value of log2 of the metabolite abundance. We have added this note to the method section.

(b) If you do a large number of pairwise tests on a data set, surely some are bound to turn out significant by just by default. Is this a problem here and later?

Response: We acknowledge that this is a possibility. However, we used a test significance threshold of α = 0.05 to minimize the possibility of false positives. Thank you for your comments.

(9) Fig 4. Networks: this may be the conventional way of presenting such results, but the figures are virtually impossible to decipher, which essentially defeats the purpose!

Response:We agree with you that the network diagram in Figure 4 contains a large amount of data that is not easy to identify. Therefore, we have replaced it with a heatmap.

Thank you very much for your professional comments, which has been extremely helpful in improving our article.

Reviewer 3 Report

Elevated temperatures by climate change could affect ecological systems in many ways. While it was reported that gut microbiota plays an important role in shifting foraging strategies of wild animals, little is known about how various biomolecules are altered during the process. In this work, Zheng Yan et al. performed a metabolomics analysis of fecal samples from giant pandas under different diets and studies the differential foraging strategies of multiple bamboo parts. The manuscript is well-structured and well-written (although some minor language curation needed) with all results clearly demonstrated and I only have a few related suggestions or comments here.

Line-40: “Ailuropoda melaneleuca” -> “Ailuropoda melanoleuca

Line-86: “a broad range of materials“ -> “a broad range of biomolecules”

Line-89: “analysis” -> “understanding”.

Line-283: Please clarify how significant differences are defined here (what comparison and thresholds are used). Although this may be mentioned in Methods already, it would be better to have them there to make the paragraph logically complete.

Figure-2 and related paragraph: the difference in metabolites and related pathways are interesting. I would suggest the authors to make comparison that would identify unique signatures for each bamboo part, for example, the authors can do (Leaf) vs. (Culm + Shoot) to identify unique metabolites enriched or depleted in Leaf.

Figure-4: (1) before performing the association studies, it would be better to have some general gut microbiota profile or analysis here. Although it was already reported in a previous publication by the authors, it would be better to have some adapted panels to recap the finding here to make a complete story. (2) Current panel-a and b are not very readable and I would suggest put make this into a network plot that microbiota taxa and all metabolites are different clusters (so the authors can also add intra-microbiota and intro-metabolites association) and use width and transparence of the links to highlight the quantitative result. (An example can be found in Figure-3 from PMID: 31591683)

Figure-S5: this figure is not very readable and please use appropriate clustering on both x and y-axis (i.e., hierarchical on y-axis and phylogenetic on x-axis) and expand the figure horizontally to make it more readable.

Moreover, I would suggest a quick additional analysis that greatly help show off the data generated by the authors: since the authors were performing 16S rRNA sequencing in a previous publication, it is difficult to get more gene-level information for pathway analysis. However, there is some existing tools (such as PICRUSt2) that the authors can use to infer how specific microbial pathway was altered with different bamboo which could provide mechanistic insight into the impact of these diet change and link them to the metabolites changes.

Author Response

Dear reviewer,

We appreciate the insightful comments from you. We have corrected the description errors and added the definition of significant difference to clarify how we defined significant differences. We have also made a comparison of unique signatures for each bamboo part. Regarding figure 4, we have added a general gut microbiota profile or analysis before performing the association studies. Furthermore, we have replaced the network diagram with a heatmap, where the microbiota taxa and metabolites are distinct clusters, to improve readability. Finally, we added the results of association analysis within gut microbial gene functions and metabolites.

Details

(1) Line-40: “Ailuropoda melaneleuca” -> “Ailuropoda melanoleuca”

Response: We appreciate your correction and have made the necessary change.

(2) Line-86: “a broad range of materials” -> “a broad range of biomolecules”

Response: Thank you for your suggestion. We have made the modification accordingly.

(3) Line-89: “analysis” -> “understanding”.

Response: We agree with your suggestion and have made the change.

(4) Line-283: Please clarify how significant differences are defined here (what comparison and thresholds are used). Although this may be mentioned in Methods already, it would be better to have them there to make the paragraph logically complete.

Response: We appreciate your suggestion and have added the definition of significant difference as follows: “variable importance prediction (VIP) score > 1 and P < 0.05”. Thank you for your suggestion.

(5) Figure-2 and related paragraph: the difference in metabolites and related pathways are interesting. I would suggest the authors to make comparison that would identify unique signatures for each bamboo part, for example, the authors can do (Leaf) vs. (Culm + Shoot) to identify unique metabolites enriched or depleted in Leaf.

Response: Thank you for your suggestion. We also considered the unique signatures for each bamboo part is important and therefore made a comparison. “The shoot group exhibited more metabolites in galactose metabolism compared to the other two groups, including galactitol, sucrose, silymarin and stachyose. The abundance of metabolites related pathways in the bamboo leaf group was not significant different compared to the other two groups simultaneously. But compared to bamboo shoot group, leaf group had significantly lower metabolites associated with the starch and sucrose metabolic pathway (including sucrose, cellobiose, and isomaltose), galactose metabolic pathway (including galactitol, sucrose, raffinose, and stachyose), pyrimidine metabolic pathway (including thymidine, uracil, uridine, and thymine, and phenylalanine, tyrosine), and tryptophan biosynthetic pathway (including quinine, shikimate, anthranilic acid (vitamin L1))”.

(6) Figure-4:

(a) before performing the association studies, it would be better to have some general gut microbiota profile or analysis here. Although it was already reported in a previous publication by the authors, it would be better to have some adapted panels to recap the finding here to make a complete story.

(b) Current panel-a and b are not very readable and I would suggest put make this into a network plot that microbiota taxa and all metabolites are different clusters (so the authors can also add intra-microbiota and intro-metabolites association) and use width and transparence of the links to highlight the quantitative result. (An example can be found in Figure-3 from PMID: 31591683)

Response: (a) We agree with you that some changes to the gut microbiota caused by bamboo food should be added here. “The diet of different bamboo parts also significantly altered the composition of the gut microbiota of giant pandas, with representative gut microbial communities for each feeding period”.

(b) Thank you very much for your suggestion. These two network plots are not very readable. Our focus here is to show the correlation between microbes and metabolites, so after discussion, we decided to replace them with a clearer heatmap.  The heatmap contains the gut microbes associated with the metabolites in the three target metabolomics and classifies the horizontal and vertical coordinates of the graph so that the results can be displayed more clearly to the reader.

(7) Figure-S5: this figure is not very readable and please use appropriate clustering on both x and y-axis (i.e., hierarchical on y-axis and phylogenetic on x-axis) and expand the figure horizontally to make it more readable.

Response: Figure S5 involves 155 gut microbes and 114 untargeted metabolites, the amount of data is too large to be readable. After discussion, we concluded that presenting the results in the form of relevant R-values and significance P-values would be more beneficial for the reader to obtain relevant data. Therefore, we organized the results by adding Table S7 “Table S7 Correlation statistics between 114 untargeted fecal metabolites and 155 gut microbes in giant pandas”.

(8) Moreover, I would suggest a quick additional analysis that greatly help show off the data generated by the authors: since the authors were performing 16S rRNA sequencing in a previous publication, it is difficult to get more gene-level information for pathway analysis. However, there is some existing too ls (such as PICRUSt2) that the authors can use to infer how specific microbial pathway was altered with different bamboo which could provide mechanistic insight into the impact of these diet change and link them to the metabolites changes.

Response: The suggestion from you is excellent. We have previously used metagenomic techniques to predict the function of the gut microbiota and found functional changes in the gut microbiota under feeding conditions in different parts of bamboo. For example, the gut microbiota of the bamboo leaf group had more enzyme gene abundance in the TCA cycle pathway (PMID: 34109451). Based on your suggestion, we made correlation analysis of these different microbial functional genes and metabolites in the present study and the results are as follows:

“In addition, we correlated the functional genes of gut microbiota with the fecal metabolites. Seven carbohydrates-related metabolites were significantly correlated with four microbial enzyme genes, with isomaltose and stachyose significantly and positively correlated with cellulose 1, 4-beta-cellobiosidase (EC 3.2.1.91). Sucrose and fructose had a significant positive correlation with bata-glucosidase (EC 3.2.1.21) (Figure 5a). Six enzyme genes of the gut microbiota involved in energy-related metabolic pathways with the highest relative abundance in the bamboo leaf group were positively correlated with lactate, GMP, pyruvate, aconitate, isocitrate and NAD, and the relative abundance of these metabolites was also highest in the bamboo leaf group (Figure 5b).”

Thank you again for you help.